# How Does Diet Change with A Diagnosis of Diabetes? Protocol of the 3D Longitudinal Study

**DOI:** 10.3390/nu11010158

**Published:** 2019-01-12

**Authors:** Emily Burch, Lauren T. Williams, Harriet Makepeace, Clair Alston-Knox, Lauren Ball

**Affiliations:** 1Menzies Health Institute Queensland, Griffith University, Gold Coast 4215, Australia; emily.burch@griffithuni.edu.au (E.B.); lauren.williams@griffith.edu.au (L.T.W.); harriet.makepeace@griffithuni.edu.au (H.M.); 2Office of the Pro-Vice Chancellor, Arts, Education and Law, Griffith University, Mount Gravatt Campus, Brisbane 4222, Australia; c.alston-knox@griffith.edu.au

**Keywords:** type 2 diabetes mellitus, nutrition, DASH, diet quality, diabetes management, dietary intake, longitudinal analysis, lifestyle management

## Abstract

Diet quality influences glycemic control in people with type 2 diabetes (T2D), impacting their risk of complications. While there are many cross-sectional studies of diet and diabetes, there is little understanding of the extent to which people with T2D change their diet after diagnosis and of the factors that impact those changes. This paper describes the rationale for and design of the 3D longitudinal Study which aims to: (i) describe diet quality changes in the 12 months following T2D diagnosis, (ii) identify the demographic, physical and psychosocial predictors of sustained improvements in diet quality and glycemic control, and (iii) identify associations between glycemic control and diet quality in the 12 months following diagnosis. This cohort study will recruit adults registered with the Australian National Diabetes Services Scheme who have been recently diagnosed with T2D. Participants will be involved in five purposefully developed telephone surveys, conducted at 3 monthly intervals over a 12-month period. Diet quality will be determined using a 24-h dietary recall at each data collection point and the data will be scored using the Dietary Approaches to Stop Hypertension (DASH) diet-quality tool. This study is the first dedicated to observing how people newly diagnosed with T2D change their diet quality over time and the predictors of sustained improvements in diet and glycemic control.

## 1. Introduction

Diet quality plays a vital role in helping people with type 2 diabetes (T2D) to achieve and maintain optimal glycemic control, thereby lowering their risk of developing diabetes-related complications [1]. Diet quality can be described as the extent to which food intake complies with national or international dietary guidelines or a priori diet quality score [2]. Investigating diet quality based on dietary patterns, defined as multiple dietary components operationalized as a single exposure [3], provides valuable information, beyond analyzing specific nutrients (e.g., protein) or food groups (e.g., dairy) [4]. This is because dietary patterns closely reflect actual dietary behavior and have a stronger influence on disease risk than specific nutrients or foods [5]. Findings from dietary pattern analyses may facilitate the translation of useful recommendations to health professionals and the general population [5,6]. A dietary pattern rich in whole-grains, fruits, vegetables, legumes, and nuts; moderate in alcohol; and low in refined grains, red or processed meats, and sugar-sweetened beverages has been shown to improve glycemic control in people with T2D [7]. Consequently, a key feature of international T2D management recommendations is to eat healthy foods that provide a high-quality diet [8,9,10].

However, evidence has shown that people with T2D have low-quality diets, despite these recommendations [10,11,12,13,14,15]. Our recent systematic review identified that internationally, people with T2D do not adhere to food group recommendations outlined in dietary guidelines [15]. Qualitative studies examining lived experiences report that people with T2D find it challenging to adopt and maintain healthy dietary behaviors after diagnosis [13,14]. Our previous qualitative study that investigated the experiences and perceptions of Australian adults newly diagnosed with T2D found that while participants reported making immediate, widespread changes to dietary behaviors that led to improvements in diet quality initially, they found it challenging to maintain dietary change [13]. Participants described feeling restricted in food choice, being uncertain of ideal dietary behaviors and felt unheard and rushed when speaking about their diet with health professionals [13]. Similar results were obtained in a qualitative study in Mexico where people reported making only short-term adherence to improvements in dietary intake due to difficulties with controlling appetite and eating with others [14]. While these qualitative findings of experiences raise concerns, it is important to also investigate quantitative aspects of diet quality change following diagnosis. 

Cross-sectional research has assessed the diet quality of people with T2D at a single-point in time [15], however, no research has quantitatively explored changes in diet quality after diagnosis. Consequently, there is no evidence as to whether diet quality remains fixed once an individual is diagnosed with T2D, or whether there are periods of marked increases or decreases in diet quality. Prospective, observational studies are valuable as they measure events in temporal sequence and can distinguish causes from effects [16,17]. Many factors influence diet quality. These include non-modifiable factors such as age and sex, and modifiable factors such as self-efficacy, perception of current diet, environmental factors such as marketing and food availability, and relationships with health professionals [11,13]. There is currently no data on the demographic and health characteristics influencing diet quality change for people with T2D [13,18]. There is a clear need to investigate how diet changes over time so targeted strategies can be developed to facilitate improved glycemic control. 

This paper describes the methodological protocol of the 3D Longitudinal Study, so named because seeing something in three dimensions adds clarity. In this case it refers to the 3D’s of Diet, after Diagnosis with Diabetes. The study aims are to: (i)Describe diet quality changes in the 12 months following T2D diagnosis. (ii)Identify the demographic, physical and psychosocial predictors of improvements in diet quality and glycemic control. (iii)Identify associations between glycemic control and diet quality in the 12 months following diagnosis.

## 2. Theoretical Framework

The ability to predict and explain health-related behavior is important for developing strategies to change those behaviors [19]. The theory of planned behavior (TPB) is among the most influential and widely applied theories of the factors influencing health-related behavior [19]. According to the TPB, the single best predictor of a person’s behavior is the intention to perform that behavior [20]. This is predicted by three constructs: attitude, subjective norm, and perceived behavioral control (PBC). The greater the PBC and more favorable the attitude and subjective norms, the stronger the intent will be to perform the behavior [20]. According to the TPB, people with T2D will intend to improve their diet quality to the extent that they believe the likely outcomes of consumption to be favorable, perceive social pressure from those who are important to them and feel capable of improving their diet quality without difficulty [21]. The constructs of the TPB are considered strong predictors of healthful eating and are commonly applied in the development of dietary behavior change interventions [21,22]. This study will integrate the TPB into its design in order to explore the factors that may serve as moderators in influencing the TPB constructs, thus affecting dietary behaviors and T2D management. 

## 3. Materials and Methods 

### 3.1. Study Design 

The 3D Longitudinal Study is a prospective observational cohort study that will be conducted in Australia between 2018–2019. The study will recruit people newly diagnosed with T2D and monitor their dietary intake over 12 months. The Strengthening the Reporting of Observational Studies in Epidemiology (STROBE) checklist for cohort studies was used to guide the development of the research protocol [23]. The 3D Longitudinal Study is registered with the Australian New Zealand Clinical Trials Registry (ANZCTR) (ref: ACTRN12618000375257) and was approved by the Griffith University Human Research Ethics Committee (ref: 2017/951). Study results will be published in peer-reviewed journals and presented at scientific conferences. 

### 3.2. Potential Participants 

Eligible participants will be adults aged 18 years or older who have been recently diagnosed with T2D (<6 months prior to recruitment contact), are registered with the Australian National Diabetes Service Scheme (NDSS) and have indicated their willingness to be contacted for research purposes. The NDSS is an initiative of the Australian Government and is administered with the assistance of Diabetes Australia [24]. In 2017, there were approximately 1.1 million people registered with the NDSS [25]. Registration is part of usual care for people diagnosed with T2D, therefore this potential participant pool provides broad representation of the target population. People with T2D are authorized to register for free if they live in Australia or are visiting from a country with which Australia has a Reciprocal Health Care Agreement on an applicable visa [24]. Registering with the NDSS enables individuals to access a range of government approved diabetes-related products and information services [24]. All registrants have the option of consenting to being contacted for research purposes. Upon registration, patients are required to register their personal details on a form signed by a registered Australian medical practitioner, nurse practitioner or a credentialed diabetes educator [24]. The majority of T2D diagnoses in Australia are made by general practitioners (GPs), who are the usual coordinators of management [7]. A detailed participant inclusion and exclusion criteria is listed in Table 1.

### 3.3. Participant Recruitment and Screening

A convenience sample of all individuals registered with the NDSS with a new diagnosis of T2D over the previous 6-month period will be sent an initial invitation letter and a plain-language summary of the research project via email by Diabetes Australia. Interested individuals will be invited to contact the research team via email or telephone to confirm eligibility, provide informed consent and arrange data collection. Participants will be informed they can withdraw from the study at any stage. This recruitment method has been trialed in a feasibility study conducted in 2016–2017 (unpublished) which successfully recruited 22 participants from 1000 email invitees. Of these 22, 17 completed baseline data collection, six participants had left the study by 3 months, however all participants remaining at 3 months were retained to 12 months.

### 3.4. Data Collection

Data will be collected using a purpose-developed, interviewer-administered telephone surveys at five-time points; baseline, and then at 3, 6, 9, and 12 months after commencing the study. Surveys will be conducted by Accredited Practicing Dietitians (APDs). The feasibility study found each survey takes approximately 30 min to complete. A measuring tape will be posted to all participants within two working days of recruitment to provide enough time to measure their waist circumference before data collection begins. Strategies including contact and scheduling methods have been shown to improve cohort retention in longitudinal studies [26]. Participants will be contacted 2 weeks prior to their next anticipated data collection round to schedule a time. All participants will be sent a reminder via their preferred contact method (email or text message) one day prior to the date of their next survey. The recruitment and contact process is outlined in Figure 1.

### 3.5. Survey Design 

Data from all secondary outcome measures will be recorded in an online survey management system: www.limesurvey.org [27]. Item wording and response options were composed to align with the Australian Bureau of Statistics (ABS) 2016 Census and the Australian Longitudinal Study on Women’s Health (ALSWH) to allow for comparison of outcomes [28,29]. Survey questions were generated using a developmental model [30] that employs five stages of questionnaire design and testing: conceptualization, design, testing, revision, and data collection. The feasibility study allowed testing of questions to ensure they were comprehendible, relevant and appropriate to participants and to confirm the survey length was suitable. Revisions were then made based on the feedback provided. For example, some participants in the feasibility study felt they were being asked the same question twice in the Healthy Eating Belief Scale. Therefore, the interviewer’s scripted introduction and description of the Healthy Eating Belief Scale was modified to notify participants that there would be some repetition. The second draft was then pilot tested on three adults outside of the research team to ensure comprehensibility, suitability and flow.

### 3.6. Outcome Measures 

Table 2 provides an overview of the primary (diet quality) and secondary outcomes and when they will be collected. 

#### 3.6.1. Primary Outcome Measure: Change in Diet Quality (measured by DASH score)

Diet quality can be measured by a variety of purpose developed tools [31]. These are constructed by assigning higher scores within sub-scales based on more frequent or higher intakes of foods, nutrients or both [31]. Dietary Approaches to Stop Hypertension (DASH) is a dietary pattern high in whole-grains, fruits, and vegetables; moderate in low-fat dairy; and low in red and processed meats, added sugars, and sodium [32]. While originally developed to assist people in the prevention and management of hypertension, DASH is now recommended for the dietary management of T2D [18,33]. Adherence to DASH positively impacts on glycemic control, weight, and hypertension, which are key indicators of risk for diabetes-related complications [5,18,32]. A randomized controlled trial (RCT) conducted in adults with T2D showed that adherence to DASH improved glycated hemoglobin (HbA1c) (−1.2%), fasting blood glucose (−0.92 mmol/L), weight (−3 kg) and waist circumference (−4.8 cm) over 8 weeks when compared with a control diet [34,35]. Those following the DASH dietary pattern, also had a greater reduction in LDL cholesterol (difference from the control diet, −7.7 ± 3.3%).

A systematic review and meta-analysis of 20 RCTs found DASH significantly reduced systolic (−5·2 mmHg) and diastolic blood pressure (−2·6 mmHg) in adults with and without diabetes [36]. Another systematic review and meta-analysis of 13 RCTs revealed that adults without T2D who adhered to DASH achieved greater weight loss (−1.42 kg), reduced Body Mass Index (BMI) (−0.42 kg/m^2^) and decreased waist circumference (−1.05 cm) compared with controls [37]. Considering the recognized impact on glycemic control, weight and hypertension, DASH was chosen as the dietary pattern used to assess diet quality in the present study. Participant DASH scores will be calculated using the DASH diet-quality tool which has been shown to have the highest correlation with health outcomes related to T2D compared to other tools that measure diet quality [38,39].

Change in DASH score from baseline to 3 months will be used to categorize participants as diet quality improvers or diet quality maintainers. Participants will be split into 2 groups; those who improved their DASH score by at least 3 DASH points (Diet quality improvers) and those who maintained their DASH score within 2.99 points or decreased their DASH score by at least 3 points (Diet quality maintainers). A change in DASH score of 3 points was selected based on findings from previous literature [40]. In a 20-year longitudinal study of over 40,000 adults, an average DASH score of 23.8 out of 40 was observed, and a change in score of approximately 3 points or more was sufficient to significantly influence long-term glycemic control [40]. 

Dietary intake data will be obtained through the Australian version of the Automated Self-Administered 24-h Dietary Assessment Tool (ASA-24). The ASA-24 is based on the validated Automated Multiple-Pass Method (AMPM) which is considered the optimal method for obtaining 24-h recall data due to its numerous probes, standardization of interviewer administration and validation against recovery biomarkers [41,42]. This method is also consistent with the methodology of the most recent population nutrition survey in Australia (the National Nutrition Survey) and has been shown to be a valid measure of dietary behavior at a given time point [43,44]. The ASA-24 is an online automated questionnaire that guides the individual through a system designed to maximize respondents’ opportunities for remembering and reporting foods eaten in the previous 24 h [45]. The questionnaire is divided into five phases in line with published methodological guidelines; ‘quick list’, ‘forgotten foods’, ‘time and occasion’, ‘detail cycle’, and ‘final probe’ [45]. These phases encourage respondents to think about their intake in different ways and from several perspectives which has been shown to reduce bias in the estimation of dietary intake [45]. Once a specific food or beverage is reported, systematic questions are asked to capture more precise information about the food, cooking methods and quantity consumed. The ASA-24, usually a self-completed tool, will be adapted for use in a telephone survey; a researcher will ask the questions and enter the data. This will reduce participant burden and help decrease any bias associated with participant information technology literacy levels. This process was carried out successfully in a feasibility study with patients newly diagnosed with T2D. The data from the feasibility study was able to be used to assess changes in diet quality over a 12-month period. 

Following data collection, participant 24-h dietary recall data will be sent from the ASA-24 program to the research team. This data will then be manually entered into FoodWorks by an experienced dietitian to allow determination of participant DASH scores. FoodWorks is a dietary analysis software program using standardized serve sizes that allows for quantification of specific food groups (e.g., vegetables) and nutrients (e.g., sodium) obtained from reported dietary intakes, recipes and meals [46]. FoodWorks draws on the national AUSNUT database [47]. AUSNUT was developed by Food Standards Australia and New Zealand and includes complete nutrient data sets of Australian foods designed specifically for nutrition surveys and is therefore suitable for use in this project [47]. 

DASH scores will be calculated using the standard scoring tool created by Fung et al [39]. Every tenth DASH score will be cross-checked by a second member of the research team to ensure accuracy. The standard scoring tool determines a score between 8 and 40 points, with 40 points representing optimal accordance with the DASH dietary pattern [39]. The DASH score is calculated by summing the number of daily servings of seven dietary components; fruits, vegetables, nuts and legumes, whole-grains, low-fat dairy, red and processed meats, added sugar, and sodium intake. For each of the components, participants are classified according to their intake ranking. Higher intakes of fruits, vegetables, low-fat dairy, whole-grains, and nuts and legumes receive higher scores. For example, quintile 1 is assigned 1 point and quintile 5 is assigned 5 points. Intake of sodium, red and processed meats and added sugars are scored in reverse as these are less desirable foods [39]. The lowest quintile is given a score of 5 points and the highest quintile is given a score of 1 point. The components scores are then summed to give an overall DASH score [39]. The scoring criteria for the DASH-style diet is outlined in Table 3.

#### 3.6.2. Secondary Outcome Measures

Secondary outcome measures will include: glycemic control, medication use, demographic factors, physical factors, psychosocial factors, and exposure to health provider support. 

##### Glycemic Control

HbA1c reflects average plasma glucose over the previous six to eight-week period [48,49]. In Australia, it is best practice for GPs to conduct HbA1c testing on people with T2D every 3 months [8]. The test is subsidized by Medicare, the Australian Government’s universal health scheme, up to four times in a 12-month period [50]. The GP on the research team will retrospectively obtain participants’ HbA1c results over the 12-month study period from the relevant pathology laboratory. Other blood results collected will include; fasting blood glucose, high-density lipoprotein cholesterol, low-density lipoprotein cholesterol, and C-reactive protein. 

##### Medication Use

Information on all medication use (name of medication and dosage) will be collected, including over-the-counter, and complementary medicines.

##### Demographic Factors 

Demographic factors collected at baseline including; age, gender, highest education level, living arrangement, self-selected social class, household income, ability to manage on income, and smoking status. Response options will be dichotomous (e.g., gender), continuous (e.g., age) or categorical (e.g., highest educational level), consistent with categories used in the national consensus by the ABS.

Age and Gender

Participants’ age at last birthday and gender will be collected. Sex response options will include three categories in line with the ABS 2016 Census; male, female and other [51]. 

Highest Education Level 

Collecting data on highest educational level helps generate a single measure of an individual’s overall educational attainment, whether it be a school or non-school qualification [51]. Participants will be asked to report their highest education level from eight categories in line with the ABS 2016 Census; postgraduate degree, graduate diploma and graduate certificate, bachelor degree, advanced diploma, diploma, certificate, year 12, or year 11 and below [51]. 

Living Arrangement 

Data on participants’ current living arrangement will be collected. In line with the ALSWH, response options will include; no one, partner/spouse, own children, someone else’s children, parents, or other adults [29]. 

Self-Selected Social Class 

Requesting information on self-selected social class allows class designation to be meaningful to participants and is more likely to reflect their actual class identity [52]. Participants will be asked to self-select their social class from one of four response options in line with the ALWHS; ‘upper class’, ‘middle class’, ‘working class’, or ‘don’t know’ [52]. 

Household Income and Ability to Manage on Income 

Gross income refers to the sum of income received from all sources before any deductions (income tax, the Medicare Levy or salary sacrificed amounts) are taken out [51]. Participants will be asked to report their average yearly gross household income from six response options in line with the ALSWH; less than $20,000, $20,001–$30,000, $30,001–$50,000, $50,001–$100,000, more than $100,000 or ‘don’t know/would rather not say’. Details on participants’ ability to manage on their current income will also be collected. Seven response options in line with the ALSWH will include; ‘impossible’, ‘difficult’, ‘always difficult’, ‘sometimes difficult’, ‘not too bad’, ‘easy’, or ‘not sure’ [53].

Smoking Status 

Participants will be asked to report their smoking status. Wording of the question has been developed to correspond with the ABS 2016 Census and response categories will include ‘yes’ or ‘no’ [51].

##### Physical Factors

Self-reported anthropometric data is valid and recommended for monitoring prevalence of obesity, particularly for large-scale studies because of its simplicity and low cost [54,55,56]. Physical factors will include; self-reported waist circumference, weight, and height. All response options will be continuous. 

Waist Circumference 

Waist circumference is a better indicator of central obesity than BMI or waist-to-hip ratio and is more strongly correlated with intra-abdominal fat content and cardiovascular risk factors [57]. Participants will be asked to self-report their waist circumference to the nearest centimeter. Participants will be asked to report results of two measures at each round of data collection, which will be averaged during data analysis. Measuring instructions adapted from the ALSWH will be provided to participants in the postal envelope [58]. Instructions will request participants to measure mid-central adiposity using the tape to measure at the level of the mid-point between the lower costal border and the iliac crest [58].

Weight

Weight will be self-reported at each data collection point to the nearest decimal point. If the participant can only report the amount in stones and pounds, a conversion factor of 2.203 will be used to convert pounds into kilograms in line with the ALSWH [52]. Participants will be asked to describe where and how their weight was measured to assess the accuracy of the information.

Height

Height will be self-reported to the nearest centimeter at baseline only. If the participant can only report the information in feet or inches, a factor of 2.54 will be used to convert inches to centimeters in line with the ALSWH [52].

BMI

BMI provides the most useful population-level measure of obesity [58,59]. Participants’ BMI will be calculated after each round of data collection using the standard equation (weight (kg)/height (m)^2^) [58]. Participants will be grouped into BMI categories for analysis according to the World Health Organization guidelines; underweight (<18.50 kg/m^2^), normal weight (18.50–24.99 kg/m^2^), overweight (≥25.00 kg/m^2^) and obese (≥30.00 kg/m^2^) [59].

Physical Activity 

Regular physical activity is a key feature of international T2D management guidelines [9,60]. This is because it has been shown to improve glycemic control (regardless of whether weight loss has occurred), lipid levels, and blood pressure in people with T2D [61,62,63]. Participants’ physical activity levels will be measured through the IPAQ-SF which is one of the most widely used physical activity assessment questionnaires and has been shown to be a valid measure of obtaining internationally comparable physical activity data [64,65]. It measures self-reported physical activity in the previous seven days and includes seven items that collect information on walking time, moderate and vigorous physical activity time and sitting time [66]. Data obtained in the IPAQ-SF will be used to estimate total metabolic equivalent (MET)-minutes per week for each participant [67].

##### Psychosocial Factors

Healthful Eating Beliefs 

Healthy eating beliefs will be investigated at each of the five data collection points to better understand their beliefs, attitudes and intentions towards making positive dietary behaviors and how these impact on diet quality change over time. Participants’ healthy eating beliefs will be assessed through the Healthful Eating Beliefs Scale. This scale is based on the TPB [68]. Standardized questions have been selected from previous Healthful Eating Beliefs Scales to suit the current study. Table 4 provides an outline of the rationale for investigation behind the questions and modes of responses. Participant scores will be generated for each of the seven subscales of the Healthful Eating Beliefs Scale by calculating the mean of the 5-point Likert scale items, with higher scores representing more positive beliefs, attitudes, and intentions towards improving dietary behaviors.

Mental Health

Mental health data will be collected at baseline, 6 and 12 months using the internationally validated K10 questionnaire [69]. This 10-item questionnaire yields a global measure of distress based on questions about anxiety and depressive symptoms experienced in the most recent four-week period [70]. It is scored using a five-level response scale based on the frequency of symptoms reported for each question. One is the minimum score for each item (none of the time) and five is the maximum score (all of the time). The maximum score is 50 indicating severe distress, the minimum score is 10 indicating no distress [70]. 

##### Exposure to GP and Allied Healthcare Support

Interactions with health professionals may help facilitate positive changes in dietary behaviors [73]. At each data collection point, participants will be asked about their concurrent and previous exposure to healthcare provider support (e.g., dietitian, diabetes educator). Participants will also be asked to report how useful they found the advice on a 5-point Likert scale (1 being ‘not at all useful’, 2 being ’slightly useful’, 3 being ‘neutral’, 4 being ‘somewhat useful’ and 5 being ‘extremely useful’). Question wording has been modified from previously conducted qualitative research that explored the experiences of dietary change in people with T2D [13].

### 3.7. Participant Confidentiality 

All study related information will be de-identified and stored securely online with password-protected access systems. Daily back-ups of all electronic data will occur to minimize any risk of lost data. Only members of the research team who need to contact study patients, enter data or perform data quality control will have access to participant information.

### 3.8. Statistical Modeling and Sample Size 

The longitudinal aspect of diet quality change will be analyzed using regression models [74]. Diet quality at 12 months (measured by the DASH score) will be the primary outcome, with glycemic control, interim diet changes (primarily at 3 months), medication use, demographic measures, physical measures, psychosocial measures, and exposure to healthcare provider support as the explanatory variables. This model will determine the relative importance of diet quality change both immediately after diagnosis and in the medium term with regards to a 12-month outcome towards sustained healthy eating. 

Sample size calculations, such as those provided in Diggle et al. [74] are not readily computed for complex regression models and assessment using simulation studies would be more appropriate [75]. However, due to the relatively small participation rate in the feasibility study (22 participants/1000 invited) credible model parameters are currently unable to be formulated, and the effect differences over time cannot be reliably inferred. Additionally, current research of diet quality change post diabetes is lacking [15], therefore effects cannot be competently elicited. As a result, rather than determining an initial sample size for the study, a Bayesian updating procedure will be used to collect data. In this manner, sample size will be calculated during the early stages of data collection to yield an idea of how many samples should be collected. Additionally, using a Bayesian framework, the parameter estimates of the mixed model will be sequentially updated as batches of data are obtained, a process known as Bayesian learning [76,77]. 

## 4. Discussion

The study described in this paper, the 3D Longitudinal Study, will be the first to observe changes in diet quality in people with T2D after diagnosis and the factors (demographic, physical, and psychosocial) that influence those changes. Longitudinal studies help highlight differences or changes in the values of one or more variables between different time periods, describe participants’ intra-individual and inter-individual changes over time and monitor the magnitude and patterns of those changes [78]. This is important for the proposed research because it is necessary to understand the extent to which people change their diet after a T2D diagnosis and why some people are able to sustain these changes over time and others are not. Understanding this will help to develop targeted strategies and facilitate enhanced dietary behavior support important to assist all people with T2D to have long-term success in improving their diet quality and help reduce the risk of complications. The results of this study will significantly add to the body of literature on the diet quality changes of people diagnosed with T2D, which is an under-researched area.

Limitations of the 3D Longitudinal Study are acknowledged. Recruitment through the NDSS is the most suitable way to access a large number of potential participants with T2D, however selection bias cannot be excluded as registration is voluntary, so participants may have greater diabetes self-management motivation. Self-reported dietary intake data and physical measurements may introduce misreporting bias and social desirability responses. Measurement errors associated with dietary assessment methods are also acknowledged. However, use of the ASA-24 h dietary recall (Australian version) which is a validated tool specifically designed for the Australian population, reduces risk of bias [43]. A feasibility study for this project has already determined that its recruitment capability, data collection and analysis procedures are achievable and appropriate. 

## Figures and Tables

**Figure 1 nutrients-11-00158-f001:**
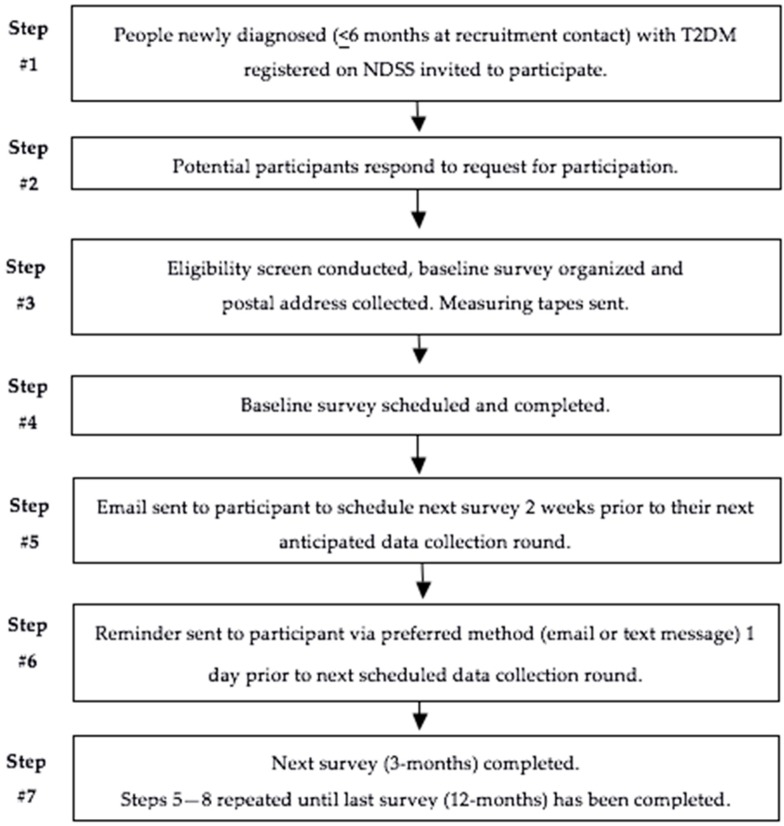
Recruitment and contact process for the 3D Longitudinal Study.

**Table 1 nutrients-11-00158-t001:** Inclusion and exclusion criteria for the 3D Longitudinal Study.

Inclusion Criteria	Exclusion Criteria
Adults aged >18 years	Individuals aged <18 years.
Diagnosed with T2D <6 months prior to recruitment contact	Diagnosed with LADA, T1D, gestational diabetes or pre-diabetes
Registered with the Australian NDSS and indicated their willingness to be contacted for research purposes.	Individuals who have been placed on a special diet due to a co-morbidity (e.g., renal disease)
Able to communicate in English	

LADA, Latent Autoimmune Diabetes in Adults; NDSS, National Diabetes Service Scheme; T1D, Type 1 diabetes; T2D, Type 2 diabetes.

**Table 2 nutrients-11-00158-t002:** Overview of data collection points in the 3D Longitudinal Study.

Data Collection Methods	Time Collected				
	0 Months	3 months	6 Months	9 Months	12 Months
Diet quality	✔	✔	✔	✔	✔
Glycemic control	✔	✔	✔	✔	✔
Medication use	✔	✔	✔	✔	✔
Baseline demographic factors	✔				
Physical factors	✔	✔	✔	✔	✔
Psychosocial factors	✔	✔	✔	✔	✔
Exposure to allied healthcare support	✔	✔	✔	✔	✔

**Table 3 nutrients-11-00158-t003:** Dietary Approaches to Stop Hypertension (DASH) dietary pattern scoring criteria.

Component	Foods	Scoring Quintiles (Q) *
Fruits	All fruits and fruit juices	Q1 = 1 point Q2 = 2 points Q3 = 3 points Q4 = 4 points Q5 = 5 points
Vegetables	All vegetables except potatoes and legumes	
Nuts and legumes	Nuts and peanut butter, dried beans, peas, tofu	
Whole-grains	Brown rice, dark breads, cooked cereal, whole-grain cereal, other grains, popcorn, wheat germ, bran	
Low-fat dairy	Skim milk, yogurt, cottage cheese	
Sodium	Sum of sodium content of all foods	Q1 = 5 points Q2 = 4 points Q3 = 3 points Q4 = 2 points Q5 = 1 point
Red and processed meats	Beef, pork, lamb, deli meats, organ meats, hot dogs, bacon	
Added sugar	Foods and beverages with added sugars (i.e., sugar sweetened beverages)	

* Q1 represents low consumption and Q5 represents high consumption.

**Table 4 nutrients-11-00158-t004:** Healthful Eating Beliefs Scale rationale for investigation and modes of responses.

Category	Rationale for Investigation	Area of Enquiry	Standardized Questions and Response Options	Source
Healthful Eating Beliefs Scale	To better understand healthy eating beliefs among participants and determine if they change over time.	Behavioral intention	“I intend to eat a healthful diet each day in the next 2 months,” 5-pt Likert “I will try to eat a healthful diet each day in the next 2 months,” 5-pt Likert “I plan to eat a healthful diet each day in the next 2 months,” 5-pt Likert	[68]
		Perceived behavioral control	“I have the self-discipline to eat a healthful diet” 5-pt Likert “I have the ability to eat a healthful diet” 5-pt Likert “Me eating a healthful diet would be easy/difficult” 5-pt Likert “Whether I do or do not follow the recommendations for my diet is entirely up to me” 5-pt Likert	[71]
		Subjective norm	“People important to me think I should not/I should eat a healthful diet” 5-pt Likert “Other people expect me to follow the daily recommendations for diet” 5-pt Likert “People important to me want me to eat a healthful diet” 5-pt Likert “Other people with diabetes follow the daily recommendations for diet” 5-pt Likert	[71,72]
		Attitudes towards self-care	“Following the recommendations for my diet would be harmful/beneficial” 5-pt Likert “It would be worthless/valuable for me to follow the daily recommendations for my diet” 5-pt Likert “Following daily recommendations for diet is unnecessary/necessary” 5-pt Likert Following daily recommendations for diet is unpleasant/pleasant” 5-pt Likert	[72]

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
