# Peer review of "How Does Diet Change with A Diagnosis of Diabetes? Protocol of the 3D Longitudinal Study"

_nutrients, 2019, doi:10.3390/nu11010158_

Round 1
Reviewer 1 Report
The reviewed manuscript describes the methodological study protocol of a prospective longitudinal study with the aim of investigating diet quality and changes over time in newly diagnosed type 2 diabetes patients. The local ethical review board has approved the study.
I have some questions that I would like to raise. The questions will not change the already approved study protocol but that may affect the way that data and study results will be handled and presented in future publications.
Dietary assessment method.
The 24-hrs recall method was chosen for the study. Can one day (24-hrs) really reflect a person’s normal diet or dietary habits? If the last 24-hrs didn’t reflect a normal day, how do the authors handle this information? Was is ever discussed to use a Food Frequency Questionnaire (FFQ), often used to measure dietary intake over a longer period (i.e. during the last month)?
Another question is regarding the 24-hrs recall software. Do the program take into account different types of fats used (butter, margarine, rapeseed- or olive oils) in recipes (in dishes reported by the participant)? How detailed is the program? Is fat quality of any interest in the study?
Collection of secondary outcomes.
To my understanding, all lab measurements (HbA1c, fasting blood glucose, HDL/LDL etc) will be collected retrospectively from different laboratories. How will the authors ensure that study participants will attend the visits at the GPs every third month according to the study protocol? My concern is about the compliance of the blood tests collected by GPs outside the study center. How will the authors handle missing data?
Self-selected social class
I do not understand the reason for collecting this information when other types of social economic data (education level, household income) already are collected? How will this variable be used, in what context?
Recruitment and sample size calculations.
It has not been described how many participants that will be enrolled to the study. It has been described that sample size calculations will be done during the early stages of the recruitment. It would have been interesting to know if the authors have any idea about how many participant they will need. It was mentioned in section 3.3 Participant recruitment, that the method on how to invite participants has been tested before. Out of 1000 invitations by email, 22 (2.2%) participants were enrolled and 11 subjects completed the study after 12 months. Based on this information, how will the authors enroll enough study participants, keep them engaged throughout the study and secure as much data (minimize missing data) to get this study as successful as it could be?
I think that when most of the variables are self-reported and collected via other instances, a large study population is necessary to see any differences between the groups of interest.
Author Response
1. The 24-hrs recall method was chosen for the study. Can one day (24-hrs) really reflect a person’s normal diet or dietary habits? If the last 24-hrs didn’t reflect a normal day, how do the authors handle this information? Was it ever discussed to use a Food Frequency Questionnaire (FFQ), often used to measure dietary intake over a longer period (i.e. during the last month)?
The 24-hour dietary recall method was chosen because it is the most common dietary intake assessment method used in food consumption surveys worldwide (1). Extensive evidence exists that has demonstrated 24-hour recalls can provide high-quality dietary intake data with minimal bias (2-4).
The ASA-24 hour dietary recall program was selected to reduce the risk of reporting bias in this study. The ASA-24 is a validated tool that has been specifically designed for the Australian population (4). The multiple‐pass 24‐hour recall method that the ASA-24 program incorporates was originally developed by the US Department of Agriculture in an effort to limit the extent of underreporting that can occur with self‐reported food intake (4). The method differs from the traditional 24‐hour recall because the interviewer uses five distinct passes to retrieve information about a participant’s food intake over the preceding 24 hours. The multiple-pass method is described in the table below:
Pass number and name | Pass details |
Pass #1, “Quick list” | Participant is asked to recall everything eaten the previous day using any recall strategy they choose. |
Pass #2, “Forgotten food list” | Interviewer probes the subject about possible forgotten foods (i.e., sweets, alcoholic beverages, sodas). |
Pass #3, “Time and occasion” | Interviewer asks standardized questions developed by the USDA to probe subjects about information pertaining to each food and clarifies food portion sizes. |
Pass #4, “Detailed cycle” | Participants are asked where the food was consumed and the length of time between eating occasions. |
Pass #5, “Final review probe” | Interviewer probes to try and get any additional food items consumed recorded as part of the recall. |
Ideally, the study would conduct repeated 24-hour dietary recalls. However, considering participants will be interviewed every 12 weeks over the space of a year, this was not considered feasible. Drop‐out rates are a prevalent complication in the analysis of data from longitudinal studies and it is believed having additional repeated recalls would dramatically increase participant burden. Many international food consumption surveys use one 24-hr recall including the: 2004 Canadian Community Health Survey, 1995 Australian National Nutrition Survey, 1998-2009 Korean National Health and Nutrition Examination Surveys and the 2008-2009 New Zealand National Nutrition Survey (1).
A FFQ was considered and rejected as not being a suitable tool for measuring short-term change in dietary intake. Most FFQs collect data that is the average intake over an extended time period such as 12 months (5). This makes them less sensitive to change in the short-term, which is vital for our study given it has 5 dietary data collection points over 12 months. We needed a tool that would be more sensitive to assessing dietary change given the main purpose of this study is to follow people longitudinally.
The 24-hour recall method is also more appropriate for this study because it allows trained Dietitians to interview participants in order to collect detailed data about food preparation methods, ingredients used in mixed dishes, and the brand names of commercial products. The use of trained Dietitians, while a more labour intensive method than an FFQ, also has the advantage of reducing the risk of misinterpretations of questions or the omission of food items that may not be understood by participants in an FFQ.
References:
1. De Keyzer W, Bracke T, McNaughton SA, et al. Cross-continental comparison of national food consumption survey methods - a narrative review. Nutrients. 2015;7(5):3587-620.
2. Kipnis V, Subar AF, Midthune D, et al. The structure of dietary measurement error: Results of the OPEN biomarker study. Am J Epidemiol. 2003;158(1):14–21.
3. Schatzkin A, Kipnis V, Carroll RJ, et al. A comparison of a food frequency questionnaire with a 24-hour recall for use in an epidemiological cohort study: results from the biomarker-based Observing Protein and Energy (OPEN) study. Intl J Epidemiol. 2003;32(6):1054–1062.
4. Moshfegh AJ, Rhodes DG, Baer DJ, et al. The US Department of Agriculture Automated Multiple-Pass Method reduces bias in the collection of energy intakes. Am J Clin Nutr. 2008;88(2):324–332.
5. Willett WC. Nutritional Epidemiology. 4th ed. New York, NY Oxford University Press, 2013.
2. Another question is regarding the 24-hrs recall software. Do the program take into account different types of fats used (butter, margarine, rapeseed – or olive oils) in recipes (in dishes reported by the participant)? How detailed is the program? Is fat quality of any interest in the study?
Yes, the ASA-24 program takes into account details on specific fats and single foods in recipes reported by participants. The recall software is extremely detailed, and the Australian version utilises the food database implemented by the Australian government in national nutrition surveys, which allows for national comparison. The software allows analysis of 41 different nutrients including: macronutrients and energy, vitamins, minerals, carotenoids, fats and cholesterol, specific fatty acids and other substances (eg. caffeine). This webpage shows the nutrient analysis available from the ASA-24 data in detail (https://epi.grants.cancer.gov/asa24/researcher/analysis.html#nutrients).
As a major macronutrient, the fat profile is of interest in the study. The benefit of having the data in a 24-hour recall format is that we not only have nutrient analysis, but can later do food group and individual food analyses according to the study outcomes.
3. To my understanding all lab measurements (HbA1c, fasting blood glucose, HDL/LDL etc) will be collected retrospectively from different laboratories. How will the authors ensure that study participants will attend the visits at the GPs every third month according to the study protocol? My concern is about the compliance of the blood tests collected by GPs outside the study centre. How will the authors handle missing data?
The 3D study is observational, rather than experimental, and the study population is located nationwide. After exploring many processes, this is the only feasible method for collecting pathology data. It will not be possible to ensure that participants attend visits to their GP every three months to have their blood tested. Rather the points of natural collection will be used. The 2016-2018 Royal Australian College of General Practitioners (RACGP) T2DM management guidelines advise Australian practitioners to collect HbA1c levels every 3-6 months, particularly during early stages of diagnosis (1). Therefore, the likelihood of missing pathology data is reduced in this specific population. If missing data does occur appropriate statistical modeling will be conducted. If the data are missing at random, linear mixed models will be our first choice. If the data are missing due to systematic reasons, the statistical analysis will be re-assessed, and appropriate survival models will be deployed.
References:
1. Royal Australian College of General Practitioners. General practice management of type 2 diabetes 2016-2018. Available from: https://static.diabetesaustralia.com.au/s/fileassets/diabetes-australia/5ed214a6-4cff-490f-a283-bc8279fe3b2f.pdf
4. I do not understand the reason for collecting this information when other types of social economic data (education level, household income) already are collected? How will this variable be used? In what context?
While we are also collecting the traditional SES measures, we acknowledge that these are only a proxy indicator of one of the key factors that influences obesity – and thereforeT2DM – that of social class. Studies are increasingly collecting this data, and will we use the data we collect to make meaningful comparisons with such studies.
The concept of social class refers to real-life groups of people, such as the working, middle and upper class, who share common class-based lifestyles, values, and identities, and may mobilise collectively as a class on political issues (1). Self-selected social class, as a subjective measure of SES, is thus likely to address facets of social identity that objective measures may not.
Social class membership is measured by allowing individuals to self-nominate to which class they perceive they belong. A number of studies have argued that subjective measures of social ranking, most often referred to in the literature as subjective social status (SSS) or subjective SES, are more significant determinants of health than objective measures of SES (2). The benefit of a subjective self-selection (self-appraisal) approach is that it reflects a socially meaningful classification (to study participants) and overcomes the limitations of assigning a social ranking based on proxy SES indicators of occupation (irrelevant for those women not employed), education (which may be misaligned to actual material living conditions), or the SES of the household or residential area of the individual (3-6). Like any measure, there are limitations of collecting data on self-selected social class, such as the possibility of misinterpretation or incorrect assignation of class membership, but in our view, this is out-weighed by the benefit of enhanced internal validity.
References
1. Connell, RW & Irving TH. 1992, Class Structure in Australian History: Poverty and Progress, 2nd edn, Longman Cheshire, Melbourne.
2. Adler NE, Epel ES, Castellazzo G, Ickovics JR. Relationship of subjective and objective social status with psychological and physiological functioning: preliminary data in healthy white women. Health Psychol. 2000;19:586-592.
3. Arber S. Comparing inequalities in women’s and men’s health. Britain in the 1990s. Social Science & Medicine. 1997;44:773–787.
4. Crompton R. (2008). Class and stratification (3rd edition). Cambridge: Polity Press.
5. Devine F, Savage M, Scott J & Crompton R. (Eds.). Rethinking class: Culture, identities and lifestyles, Houndmills: Palgrave Macmillan.
6. Krieger N, Williams D, & Moss N. Measuring social class in US public health research. Concepts, methodologies, and guidelines. Annual Review of Public Health. 1997;18:341–378.
5. It has not been described how many participants that will be enrolled to the study. It has been described that sample size calculations will be done during the early stages of the recruitment. It would have been interesting to know if the authors have any idea about how many participants they will need. It was mentioned in section 3.3 Participant recruitment, that the method on how to invite participants has been tested before. Out of 1000 invitations by email, 22 (2.2%) participants were enrolled and 11 subjects completed the study after 12 months. Based on this information, how will the authors enrol enough study participants, keep them engaged throughout the study and secure as much data (minimize missing data) to get this study as successful as it could be? I think that when most of the variables are self-reported and collected via other instances, a large study population is necessary to see any difference between the groups of interest.
The statistical models being used in this analysis are regression based, and as such, there are no straight forward sample size calculations. For this reason, we will employ a Bayesian updating strategy and forecast required sample size during the course of the study. During the course of the study we will use current responses to simulate scenarios and sample size/power accordingly. We have a paragraph which highlights why a sample size calculation is not appropriate for the regression models included in this study:
“Sample size calculations, such as those provided in Diggle et al [74] are not readily computed for complex regression models and assessment using simulation studies would be more appropriate [75]. However, due to the relatively small participation rate in the feasibility study, credible model parameters are currently unable to be formulated, and the effect difference over time cannot be reliably inferred. Additionally, current research of diet quality change post diabetes is lacking [15], therefore effects cannot be competently elicited. As a result, rather than determining an initial sample size for the study, a Bayesian updating procedure will be used to collect data. In this manner, sample size will be calculated during the early stages of data collection to yield an idea of how many samples should be collected. Additionally, using a Bayesian framework, the parameter estimated of the mixed model will be sequentially updated as batches of data are obtained, a process known as Bayesian learning [76, 77]” (Methods: page 11, lines 370-380).
Specific methods will be employed in attempt to engage participants and reduce drop-out rates in this study. For example, the same Dietitian will conduct participant interviews at each data collection point to allow for rapport building and gift voucher incentives for participation will be provided.
The feasibility study had a high retention rate at 12 months, which suggests participants will remain engaged. We have outlined this in the following paragraph:
“This recruitment method has been trialled in a feasibility study conducted in 2016 – 2017 (unpublished) which successfully recruited 22 participants from 1000 email invitees. Of these 22, 17 completed baseline data collection, six participants had left the study by 3 months, however all participants remaining at 3 months were retained to 12 months” (Methods: page 3, lines 130-133).

Reviewer 2 Report
The current protocol describes the rationale for and design of the 3D Longitudinal study, which is a cohort study with the following aims 1) describe diet quality changes in the 12 month after type 2 diabetes diagnosis, 2) identify demographic, physical and psychosocial predictors of sustained improvements in diet quality and glycaemic control, 3) identify associations between glycaemic control and diet quality in the 12 months following diagnosis. This will be done by the use of telephone interviews over a 12 month period and the Dietary Approaches to stop Hypertension (DASH) diet quality tool. The rationale and design is well described in this protocol, and the proposed cohort study is relevant.
Comments:
Regarding the focus on glycaemic control, the ACCORD study demonstrated that type 2 diabetes patients who were tightly regulated in terms of HBa1c had increased mortality and did not experience significantly reduced major cardiovascular events. Could the authors please comment on these results in relation to the planned 3D Longitudinal study? ACCORD study:
Effects of Intensive Glucose Lowering in Type 2 Diabetes. The Action to Control Cardiovascular Risk in Diabetes Study Group. N Engl J Med. Author manuscript; available in PMC 2015 Aug 27. Published in final edited form as: N Engl J Med. 2008 Jun 12; 358(24): 2545–2559. Published online 2008 Jun 6. doi: 10.1056/NEJMoa0802743
Page 3, table 1. Please consider adding LADA as exclusion criteria.
Page 5, line 172. Please consider mentioning that the RCT found that adherence to DASH also improved the cholesterol levels among others.
Page 5, line 177. A word is missing after recognized.
Page 6, table 3. The scoring quintiles seem to be incorrectly inserted in the table.
Page 7, line 240. General practitioners (GPs) was also defined on page 3, line 116.
Page 9, table 4. “Me instead of “My eating a healthful diet”.
Page 10, table 4, last sentence. Uppercase U in “Unpleasant”.
Author Response
1. Regarding the focus on glycaemic control, the ACCORD study demonstrated that type 2 diabetes patients who were tightly regulated in terms of HBa1c had increased mortality and did not experience significantly reduced major cardiovascular events. Could the authors please comment on these results in relation to the planned 3D Study? ACCORD study: Effects of Intensive Glucose Lowering in Type 2 Diabetes. The Action to Control Cardiovascular Risk in Diabetes Study Group. N Engl J Med. Author manuscript: available in PMC 2015 August 27. Published in final edited form as: N Engl J Med. 2008 June 12; 358(24): 2545-2559. Published online 2008 June 6. Doi: 10.1056/NEJMoa0802743.
The authors agree that the findings of the ACCORD study are important. However, the findings are not relevant to the aims of the observational 3D study or the specific 3D study participant group. The 3D study is purely observational and is not conducting any intervention that may put patients at risk by enforcing intense glycaemic control strategies. The studies differ in the following ways:
1. They have different populations: The ACCORD trial enrolled high risk patients with T2DM who had been treated an average of 10 years at the time of enrollment. The ACCORD participant inclusion criteria was that participants needed to have either a previous CVD event or at least two CVD risk factors (please see inclusion/exclusion criteria here https://clinicaltrials.gov/ct2/show/NCT00000620). This suggests that the results of the study are not necessarily generalizable to all patients with T2DM, particularly those early in the course of their disease. The 3D study is focused on newly diagnosed patients (those diagnosed in the previous 6 months or less at the time of recruitment).
2. They have different study designs: the ACCORD trial was a randomized trial comparing intensive therapy with standard care. As such, the aims also vary, given the 3D study is an observational design that aims to: (i) describe diet quality changes in the 12 months following T2D diagnosis, (ii) identify the demographic, physical and psychosocial predictors of sustained improvements in diet quality and glycemic control, and (iii) identify associations between glycemic control and diet quality in the 12 months following diagnosis. While several dietary intervention studies have looked at effect on glycaemic control, there are no longitudinal studies of dietary change. So, this will be the first study to assess the longitudinal relationship between glycaemic control and dietary change observationally.
2. Page 3, table 1. Please consider adding LADA as exclusion criteria.
Table 1, page 3 has now been updated to include the following sentence under the ‘Exclusion criteria’.
“Diagnosed with LADA, T1D, gestational diabetes or pre-diabetes”.
3. Page 5, line 172. Please consider mentioning that the RCT found that adherence to DASH also improved the cholesterol levels among others.
Methods section has been updated to read:
“A randomized controlled trial (RCT) conducted in adults with T2D showed that adherence to DASH improved glycated hemoglobin (HbA1c) (−1.2%), fasting blood glucose (−0.92 mmol/L), weight (−3 kg) and waist circumference (−4.8 cm) over 8 weeks when compared with a control diet [34, 35]. Those following the DASH dietary pattern, also had a greater reduction in LDL cholesterol (difference from the control diet, -7.7 + 3.3%).” (Methods: page 5, lines 171-173)
4. Page 5, line 177. A word is missing after recognized.
The sentence has been updated to include the missing word:
“Considering the recognized impact on glycemic control, weight and hypertension, DASH was chosen as the dietary pattern used to assess diet quality in the present study.” (Discussion: page 5, line 180-182)
5. Page 6, table 3. The scoring quintiles seem to be incorrectly inserted in the table.
The scoring quintiles are correctly inserted in Table 3, page 6. Table 3 has been re-arranged to include a separation line between the two groups and numbering for each food group component to make it clearer to the reader. To clarify:
- The top section (components 1-5) are the more desirable food groups in the scoring criteria. The scoring quintiles in the column on the right hand side of the table show that those who are in the highest quintile (quintile #5) for these food groups, score the highest score of 5 points.
- The bottom section (components 6-8) are the less desirable food groups in the scoring criteria. The scoring quintiles in the column on the right hand side of the table show that those who are in the highest quintile (quintile #5) for these food groups, score the lowest score of 1 point.
6. Page 7, line 240. General practitioners (GPs) was also defined on page 3, line 116.
This has been amended. The “GP” abbreviation is now defined only on page 3, line 116.
7. Page 9, table 4. “Me instead of “My eating a healthful diet”.
The text has been amended in the manuscript to read:
“Me eating a healthful diet.” (Methods: page 10, Table 4).
8. Page 10, table 4, last sentence. Uppercase U in “Unpleasant”.
The text has been amended in the manuscript to read:
“Following daily recommendations for diet is unpleasant/pleasant.” (Methods: page 10, Table 4).
